# Kölliker–Fuse/Parabrachial Complex PACAP—Glutamate Pathway to the Extended Amygdala Couples Rapid Autonomic and Delayed Endocrine Responses to Acute Hypotension

**DOI:** 10.3390/ijms262311405

**Published:** 2025-11-25

**Authors:** Vito S. Hernández, Pedro Segura-Chama, Limei Zhang

**Affiliations:** Department of Physiology, School of Medicine, National Autonomous University of Mexico, Mexico City 04510, Mexico; vitohdez@unam.mx (V.S.H.); segurapd@gmail.com (P.S.-C.)

**Keywords:** PACAP (pituitary adenylate cyclase-activating polypeptide), Kölliker–Fuse nucleus, parabrachial complex, extended amygdala, PKCδ neurons, GluD1 receptor, calyx-like synapse, autonomic regulation, neuroendocrine control, Fos expression

## Abstract

The calyx of Held is a giant axo-somatic synapse classically confined to the auditory brainstem. We recently identified morphologically similar calyx-like terminals in the extended amygdala (EA) that arise from the ventrolateral parabrachial complex and co-express PACAP, CGRP, VAChT, VGluT1, and VGluT2, targeting PKCδ^+^/GluD1^+^ EA neurons. Here, we asked whether this parabrachial–EA pathway participates in compensation during acute hypotension. In rats given hydralazine (10 mg/kg, i.p.), we quantified Fos protein during an early phase (60 min) and a late phase (120 min). Early after hypotension, Fos surged in a discrete subpopulation of the parabrachial Kölliker–Fuse (KF) region and in the EA, whereas magnocellular neurons of the supraoptic and paraventricular nuclei (SON/PVN) remained largely silent. By 120 min, magnocellular SON/PVN neurons were robustly Fos-positive. Confocal immunohistochemistry showed that most Fos+ PKCδ^+^/GluD1^+^ EA neurons were encircled by PACAP+ perisomatic terminals (80.8%), of which the majority co-expressed VGluT1 (88.1%). RNAscope in situ hybridization further identified a selective KF population co-expressing *Adcyap1* (PACAP) and *Slc17a7* (VGluT1) that became *fos*-positive during the early phase. Together, these data suggest that a KF PACAP^+^/VGluT1^+^ projection forms calyceal terminals around PKCδ^+^/GluD1^+^ EA neurons, providing a high-fidelity route for rapid autonomic rebound to falling blood pressure, while slower endocrine support is subsequently recruited via neurohormone-magnocellular activation. This work links multimodal parabrachial output to temporally layered autonomic–neuroendocrine control.

## 1. Introduction

The extended amygdala (EA), comprising the capsular division of the central amygdala (CeC) and the oval nucleus of the bed nucleus of the stria terminalis (BSTov), is a hub for rapid viscerosensory integration and for coordinating autonomic, behavioral, and emotional responses to threats and homeostatic challenges [1,2]. These nuclei receive dense inputs from brainstem centers that relay visceral and nociceptive signals—most notably the parabrachial nucleus (PBN)—which alert the EA to internal bodily states and imminent threats [3,4]. In turn, the CeC and BSTov project to hypothalamic and medullary effectors to orchestrate defensive and stress-related responses; for example, the central amygdala modulates freezing, startle, and autonomic adjustments, including heart-rate and blood pressure changes, via projections to the rostral ventrolateral medulla (RVLM) and hypothalamus [5,6,7,8,9]. Rapid excitation of CeC neurons, which disinhibit downstream sympatho-excitatory pathways, is therefore well positioned to influence blood pressure control.

We recently identified a multimodal calyceal synapse in the EA [10] formed by PACAP-expressing neurons of the Kölliker–Fuse (KF) nucleus, a ventrolateral subdivision of the parabrachial complex (PBc) [10,11,12]. These giant axo-somatic terminals co-package glutamate, acetylcholine, and multiple neuropeptides, enveloping PKCδ^+^/GluD1^+^ neurons in the CeC and BSTov. Confocal microscopy, focused-ion-beam scanning electron microscopy (FIB-SEM), and 3-D reconstructions confirmed multiple synaptic specializations onto this PKCδ subpopulation. The terminals co-express VGluT1, VGluT2, VAChT, PACAP, CGRP, neurotensin, and calretinin, whereas the postsynaptic targets express GluD1, a synaptic adhesion molecule implicated in stabilizing large glutamatergic contacts [13,14], suggesting convergence of excitatory, cholinergic, and peptidergic signaling at a single high-fidelity axo-somatic interface [15].

After the discovery of the KF/PBc complex-to-extended amygdala (EA) high-fidelity (HiFi) pathway, an immediate question arose regarding its physiological function. Although the PBc has been extensively studied for its roles in pain modulation and fear responses [16,17], the presence of such a high-fidelity and metabolically demanding pathway implies its participation in homeostatic mechanisms that operate continuously under normal conditions and are rapidly recruited by frequent or even subtle stressors that disturb internal equilibrium.

Located at the pontine–mesencephalic junction, the Kölliker–Fuse (KF) nucleus has long been associated with respiratory phase control; however, converging evidence now places the KF as a key cardiorespiratory and autonomic node that can influence sympathetic outflow and arterial pressure [18,19,20,21]. In rats, KF neurons recruit post-inspiratory vagal and sympathetic nerve activities during hypoxemic stimuli and engage sympathetic chemoreflex pathways, linking this nucleus directly to blood pressure regulation [19]. In parallel, parabrachial/KF outputs provide salient interoceptive drive to the extended amygdala, where hypotension activates parabrachial to CeA neurons, and CeA neurons, in turn, project monosynaptically to RVLM sympathetic premotor neurons, mechanistic routes through which KF signals can influence blood pressure [3,7].

Here, using a hydralazine-induced acute hypotension model in rats, we show that the KF to CeC/BSTov calyceal pathway is rapidly recruited during the early phase of blood pressure recovery, independently of hypothalamic magnocellular activation. We propose a two-tiered compensation: a fast, high-fidelity KF to EA neurotransmission that reinstates blood pressure, followed by slower hypothalamo–neurohypophysial support that sustains blood pressure stabilization.

## 2. Results

Pioneering work had hinted that the parabrachial complex–to–CeC pathway is recruited during acute hypotension [3]. To evaluate the physiological significance of the Kölliker–Fuse (KF) to the extended amygdala (EA) pathway, we employed the hydralazine (HDZ)-induced acute hypotension model [22] for probing this pathway’s involvement in blood pressure recovery.

Systemic administration of hydralazine (10 mg/kg, i.p.) produced a rapid and marked fall in arterial blood pressure (BP) within the first few minutes after injection (Figure 1B–D). The mean arterial pressure (MAP) decreased from 112.8 ± 1.7 mm Hg at baseline to 82.8 ± 3.9 mm Hg at 5 min and reached 68.9 ± 6.4 mm Hg at 10 min, followed by a gradual recovery phase in which the difference from saline controls was no longer significant by 90 min (98.2 ± 6.9 mm Hg). A concurrent transient tachycardic response was observed: baseline heart rate was 389.7 ± 2.6 BPM at time 0 and increased significantly to 439.9 ± 10.4, 413.7 ± 3.9 and 421 ± 12.6 BPM at 5, 10 and 15 min after HDZ injection, respectively (* *p* < 0.05, ** *p* < 0.01, Figure 1A). Appendix A shows the detailed heart rate measurements at each timepoint.

To document interindividual variability and ensure the robustness of the group means, three consecutive BP measurements per animal were obtained at each time point (Appendix A, and the corresponding individual MAP trajectories are displayed in Appendix A. These data confirm that all HDZ-treated animals exhibited a characteristic hypotensive phase followed by partial or complete recovery within 120 min, whereas saline-treated controls maintained stable BP values throughout the experiment.

With the acute HDZ hypotensive effect validated, we sought to identify the brain regions engaged during the compensatory response to hypotension. It is well established that c-*fos* mRNA is induced within minutes of cellular activation (≈5–20 min), and that c-Fos protein becomes reliably detectable by immunohistochemistry starting at ~20–30 min and up to 60–90 min post-stimulus [23,24]. we mapped Fos (protein) immunoreactivity at 60 min (“early-Fos”) and 120 min (“late-Fos”) after HDZ injection (Figure 2A). Perfused-fixed brain at 60 min of HDZ challenge, a distinct subpopulation within the ventrolateral parabrachial complex, within the Kölliker–Fuse nucleus (KF), showed prominent Fos induction, together with strong labeling in the capsular central amygdala (CeC) and oval BNST (BSTov) (Figure 2(B3–D3)). In contrast, magnocellular neurons of the supraoptic (SON) and paraventricular (PVN) nuclei remained largely quiescent at this time point (Figure 2(E3)). By 120 min, however, the SON and PVN exhibited dense nuclear Fos expression (Figure 2(E4)), marking a late phase of activation. Quantitative analyses confirmed a significant increase in Fos-positive cell density in KF, CeC, and BSTov at 60 min time point, followed by pronounced labeling in PVN/SON at 120 min (Figure 2F; one-way ANOVA, different letters indicate *p* < 0.05). These results delineate a two-stage temporal pattern: an early activation of the KF to EA network that precedes a later recruitment of hypothalamo-neurohypophysial neurons responsible for neurohormone release.

To identify the cellular targets recruited during the early compensatory phase of HDZ-induced hypotension, we combined Fos immunofluorescence with markers of the calyceal input previously characterized in the extended amygdala [15]. As c-*fos* mRNA is induced within minutes of neuronal activation and the Fos protein becomes detectable by immunohistochemistry within 30–90 min [23], Fos labeling at 60 min post-HDZ injection was interpreted as reflecting neurons activated during the initial 5–15 min following the hypotensive stimulus.

At 60 min post-HDZ, numerous Fos-positive nuclei were observed within the capsular central amygdala (CeC), where they were surrounded by ring-like PACAP-immunoreactive terminals (Figure 3A). High-resolution thin slice (<1 µm, 1 airy under 63× objective) confocal imaging revealed that these perisomatic rings corresponded to VGluT1^+^/PACAP^+^ axosomatic terminals apposed to PKCδ^+^ neurons (Figure 3B). Many of these terminals also contained CGRP and were positioned adjacent to GluD1-positive postsynaptic puncta (Figure 3C,D).

A similar pattern was found in the oval subdivision of the bed nucleus of the stria terminalis (BSTov), where Fos-positive neurons were surrounded by PACAP-immunoreactive calyceal structures, many of which co-expressed VGluT1 and contacted PKCδ^+^ somata (Figure 3E–H). Quantitative analysis showed that 80.8% of Fos-positive neurons were encircled by PACAP^+^ calyceal terminals (Figure 3I), and 88.1% of these perisomatic rings co-expressed VGluT1 (Figure 3J).

Electron microscopic micrographs showing the calyceal synaptic structures with PACAP, VGluT1/2 in the presynaptic component and GluD1 and PKCδ in the postsynaptic cell body are shown in the accompanying paper [10].

These findings indicate that the KF to EA HiFi pathway previously described with peculiar characteristics of PACAP/CGRP/VGluT1-positive calyceal input to PKCδ/GluD1-expressing CeC and BSTov neurons is selectively engaged during the early homeostatic response to acute hypotension.

To identify the upstream neurons giving rise to the PACAP/VGluT1 calyceal terminals, we examined gene expression in the KF using single- and duplex-RNAscope in situ hybridization. *Adcyap1* (PACAP mRNA)-positive neurons formed a dense cluster within the ventrolateral parabrachial complex (Figure 4A), and many also expressed *Slc17a7* (VGluT1 mRNA; Figure 4B,E). Duplex assays combining Fos with *Adcyap1* or *Slc17a7* revealed that a subpopulation of PACAP^+^/VGluT1^+^ KF neurons became Fos-positive at 60 min following HDZ (Figure 4C,D). Additional duplex reactions confirmed partial co-expression of *Adcyap1* and *Calca* (CGRP) (Figure 4F), consistent with the mixed neurochemical profile of the calyceal terminals in the CeC. Together, these data identify a discrete PACAP/VGluT1-expressing KF subpopulation that is activated during the early phase of hypotensive challenge and likely constitutes the source of the PACAPergic calyceal projection to the extended amygdala.

In order to determine whether individual KF neurons could provide direct anatomical projections to the extended amygdala regions that receive PACAP-positive calyceal terminals identified in the present study, as well as in a recent study [15], we used the in vivo juxtacellular labeling method [25,26]. We performed 14 independent attempts. Eight penetrations were located outside the KF region, as confirmed histologically. Among the six recordings within the KF, by locating the glass electrode tips, two neurons were unlabeled, and the other three showed incomplete filling, with only somatic and proximal dendrites staining. A single neuron was found with soma located within the KF subnucleus and was extensively labeled (Figure 5), displaying well-preserved morphology of dendrites traceable across more than six consecutive parasagittal sections surrounding the soma. The axon of this neuron entered the superior cerebellar peduncle (SCP), a major white matter tract that conveys fibers toward the diencephalon. Although the axon trajectory could not be continuously followed within the SCP due to fiber density, labeled axon terminals were detected in both the capsular central amygdala (CeC) and the oval subdivision of the bed nucleus of the stria terminalis (BSTov).

This juxtacellularly labeled KF neuron thus provides direct anatomical evidence linking a pontine source to the extended amygdala targets receiving PACAP-rich calyceal terminals. Despite the low yield inherent to in vivo labeling of small pontine nuclei, the preserved morphology and projection pattern strongly support the existence of a monosynaptic KF to EA pathway. Together with the Fos activation data, these findings consolidate the view that the KF to EA circuit constitutes a functional conduit for autonomic and homeostatic information transfer during acute hypotensive stress.

## 3. Discussion

Acute hypotension elicits a rapid autonomic rebound followed by slower endocrine support. Here we identify a discrete Kölliker—Fuse (KF) PACAP/VGluT1 pathway to the extended amygdala (EA) that is engaged during the early phase of blood pressure recovery, preceding hypothalamo–neurohypophysial activation. Our findings reveal a previously unrecognized synaptic mechanism that couples a rapid autonomic response with a delayed neuroendocrine compensation after acute hypotension. The identification of a PACAP/VGluT1/CGRP calyceal projection from the Kölliker—Fuse (KF) nucleus to PKCδ^+^/GluD1^+^ neurons of the extended amygdala (EA) introduces a new element into the organization of central networks that control blood pressure. This discovery expands the classical model of cardiovascular homeostasis beyond medullary baroreflex circuits, incorporating a high-fidelity pontine—amygdaloid route capable of influencing sympathetic outflow with exceptional temporal precision.

The biphasic Fos pattern observed, i.e., early activation in the KF and EA followed by delayed recruitment of vasopressinergic magnocellular neurons, suggests a two-stage temporal hierarchy in the brain’s response to falling blood pressure. In the first stage, PACAP/VGluT1/CGRP calyceal input rapidly excites PKCδ^+^ neurons in CeC and BSTov, which are strategically positioned to modulate sympathoexcitatory outputs through disinhibition of hypothalamic and medullary targets [2,27]. This fast relay may provide the neural substrate for the “autonomic rescue” phase, restoring arterial pressure before endocrine mechanisms become engaged. The functional organization of the CeA microcircuit offers a plausible mechanism by which KF input could translate into sympathoexcitatory drive. PKCδ^+^ neurons in the central capsular subdivision (CeC/CeL) provide inhibitory control over output neurons in the medial CeA (CeM), which themselves project to the rostral ventrolateral medulla (RVLM) and other premotor autonomic nuclei [8,28]. Activation of PKCδ^+^ neurons in the CeC by PACAP/VGluT1 calyceal input could functionally reduce inhibitory drive from CeM output neurons [29] to the rostral ventrolateral medulla (RVLM), thereby facilitating sympathetic rebound during acute hypotension. We, therefore, propose that activation of PKCδ^+^ neurons in the CeC by PACAP/VGluT1 calyceal input may participate in a disinhibitory mechanism that ultimately facilitates RVLM sympathetic premotor activity. This working hypothesis is based on known CeA microcircuit architecture [29] rather than direct evidence obtained here, and should be tested in future experiments using targeted tracing or electrophysiological approaches.

In the context of acute hypotension, PACAP/VGluT1 calyceal input from the KF to PKCδ^+^ CeC neurons could therefore reduce CeM output, disinhibiting RVLM sympathetic premotor neurons and facilitating the rapid restoration of arterial pressure. This interpretation reconciles the inhibitory nature of PKCδ^+^ neurons with the net excitatory outcome on sympathetic tone, linking the KF to EA projection anatomically and functionally to the baroreflex recovery phase.

The identification of PKCδ^+^ neurons as principal recipients of KF input is consistent with their well-established role as inhibitory gatekeepers within the central amygdala microcircuitry. A previous study [29] used optogenetic tools to demonstrate that PKCδ^+^ neurons in the lateral/central amygdala (CeL/CeC) exert potent inhibitory control over medial output neurons (CeM), forming a disinhibitory switch that governs behavioral and autonomic responses to salient stimuli. Within this framework, the KF PACAP/VGluT1 calyces described here could represent a high-fidelity excitatory drive onto these inhibitory gate neurons, biasing the circuit toward rapid activation of downstream sympathoexcitatory pathways when visceral homeostasis is threatened. Such an arrangement would allow the extended amygdala to function as a dynamic relay, integrating pontine visceral inputs with affective context while gating the flow of information to medullary autonomic centers. Thus, our findings extend the principle that PKCδ^+^ neurons act as modulatory nodes beyond conditioned fear paradigms to the realm of cardiovascular control, highlighting their broader role in orchestrating state-dependent autonomic output.

The transient tachycardia observed during the early phase of blood pressure recovery (Figure 1A) is consistent with a baroreflex-mediated compensatory response in HDZ challenge [21]. The rapid fall in arterial pressure following hydralazine injection likely triggered inhibitory input from barosensitive neurons of the nucleus tractus solitarius (NTS) to medullary depressor circuits, resulting in reflex sympathetic activation and heart rate acceleration [21]. The concomitant Fos induction within the KF and extended amygdala at 60 min suggests that these higher-order centers may participate in sustaining or amplifying this autonomic rebound once the initial brainstem reflex has been engaged. In this context, excitation of PKCδ^+^ neurons in the CeC and BSTov by PACAP/VGluT1 calyceal input could modulate descending sympatho-excitatory pathways through disinhibition of medullary targets, thereby reinforcing the baroreflex-driven tachycardia and facilitating blood pressure restoration. Although direct sympathetic recordings were not performed, this interpretation aligns with established models of integrated cardiovascular compensation linking parabrachial, amygdalar, and medullary circuits.

The second delayed stage engages magnocellular SON/PVN neurons to ensure sustained vasopressin-mediated volume recovery and allostatic stabilization [30]. The delayed recruitment of magnocellular neurons in the PVN and SON likely reflects the operation of local circuits that gate hypothalamic output during stress and autonomic challenges. As proposed by Herman et al. (2002), the PVN functions not as a simple relay but as a microcircuit hub where excitatory and inhibitory inputs from limbic and brainstem sources converge onto GABAergic and glutamatergic interneurons before influencing neurosecretory and preautonomic neurons [31]. Such organization provides a mechanism for temporal filtering, in which rapid autonomic inputs, potentially relayed through the KF—EA pathway, activate upstream excitatory drives that are initially counterbalanced by strong local inhibition. Only when this inhibitory tone is lifted, or when excitatory inputs summate beyond a critical threshold, do PVN magnocellular neurons become engaged, producing the late phase of Fos induction and vasopressin release. This ensures that neuroendocrine compensation occurs only after the initial autonomic rebound has stabilized arterial pressure, thereby preventing redundant or excessive activation of vasopressinergic systems.

Converging anatomical evidence reinforces the notion that the extended amygdala exerts direct influence over medullary sympathetic control centers. For instance, anterograde tracing from the rat central amygdala (CeA) combined with electron microscopy demonstrated a direct CeA → RVLM projection making synaptic contacts onto phenylethanolamine-N-methyltransferase–positive (C1) medullary sympathetic premotor neurons activated by hypotension [7]. Using cholera toxin beta, CeA afferents to the ventrolateral medulla were confirmed, further supporting a continuous extended-amygdala → VLM/RVLM axis. These studies positioned the CeA as an efferent component of the cardiovascular reflex integration. Also, it has been shown that somatostatin-containing projections from CeA also reach the RVLM, indicating the existence of a heterogenic amygdalo-medullary pathway that potentially has effects of modulating the sympathetic tone [8,28].

Additionally, it has been shown In the cat that the BNST also sends descending fibers to ventrolateral medullary autonomic fields including the lateral tegmental field, nucleus of the solitary tract, and dorsal vagal complex, overlapping ventrolateral medullary regions that flank the RVLM, also the authors discuss the similarity of the terminal fields innervated by the CeC and the BNST, suggesting the possibility to consider both regions as aone anatomical entity [6].

The calyx-like terminals described here resemble those of the auditory brainstem in scale and transmitter complexity, but they occur in a forebrain circuit governing visceral/autonomic state rather than sound localization. Their morphology and multi-transmitter composition imply exceptionally reliable transmission with the potential for graded modulation by neuropeptides. PACAP, co-released with glutamate, likely enhances presynaptic Ca^2+^ influx and postsynaptic excitability, thus amplifying the gain of the autonomic response [11]. The co-presence of VAChT in these PACAP/CGRP terminals suggests synergistic recruitment of peptidergic and cholinergic modulation, conferring both precision and persistence. Functionally, this arrangement may enable the EA to detect visceral perturbations not merely as sensory inputs but as urgent salience signals requiring rapid autonomic adjustment.

The juxtacellular reconstruction of an individual KF neuron provides direct anatomical continuity from the pontine source to forebrain targets: a rostrally directed axon traveling through the SCP and dorsal tegmental bundle to reach the CeC and then the BSTov via the ansa peduncularis, with calyceal-like terminals observed in BSTov and PACAP-apposed terminals in CeC. This single-cell evidence dovetails with our RNAscope and confocal findings—PACAP/VGluT1 KF neurons activated early (Figure 4) and PACAP/VGluT1 calyces encircling Fos+ PKCδ/GluD1 EA neurons (Figure 3)—thereby closing the loop from molecular identity → activation → pathway anatomy → target engagement. Together with the sequential Fos mapping (Figure 2), the data support a model in which a high-fidelity KF→EA channel initiates rapid autonomic recovery, while magnocellular SON/PVN neurons provide delayed endocrine stabilization.

By positioning the EA as a recipient of KF autonomic drive, our results bridge two traditionally distinct domains: viscerosensory control and emotional processing. The same PKCδ^+^ EA neurons involved in hypotensive compensation also regulate fear and anxiety states [32,33,34]. Thus, the calyceal pathway could represent an anatomical substrate through which internal bodily challenges engage affective circuits, giving rise to the subjective awareness of “bodily threat” [35]. The dual role of PACAP in both stress [36,37] and cardiovascular regulation [38] supports this idea: PACAP signaling in the amygdala and BNST has been implicated in anxiety and panic disorders [39,40]. Our data therefore offer a mechanistic framework for understanding how visceral disturbances can trigger emotional arousal, unifying autonomic and affective physiology.

### Limitations and Future Directions

While the calyceal morphology and transmitter composition strongly suggest a high-fidelity mode of transmission, our conclusions remain primarily anatomical and correlative. Direct physiological evidence of fast excitatory postsynaptic currents (EPSCs) at these KF to EA terminals is still lacking. Future ex vivo or in vivo patch-clamp recordings from PKCδ^+^/GluD1^+^ neurons in the CeC and BSTov, combined with optogenetic or chemogenetic activation of KF PACAP/VGluT1 neurons, will be required to measure EPSC kinetics and verify high-fidelity transmission. Likewise, continuous blood pressure telemetry and targeted disruption of PACAP or VGluT1 signaling could determine the causal contribution of this pathway to autonomic recovery. Finally, quantifying circulating or cerebrospinal catecholamines and vasopressin at successive post-hypotension intervals would complement the current Fos-based approach and clarify how the rapid synaptic and slower endocrine phases integrate over time [3].

This study introduces the first direct evidence of a multimodal calyceal synapse operating within a limbic–autonomic network. Beyond hypotension, such architecture may contribute to cardiorespiratory coupling during hypoxia [19] or to stress-induced sympathetic activation [41]. Future electrophysiological work would be key to determining whether PACAP-dependent facilitation at these terminals shows activity-dependent plasticity, potentially explaining persistent sympathetic bias in chronic stress or hypertension. Likewise, identifying homologous KF to EA pathways in humans [42] may clarify the neural basis of dysautonomic states and anxiety-related cardiovascular symptoms.

## 4. Materials and Methods

Adult (>60 days) male Wistar rats were obtained from the vivarium of the School of Medicine, National Autonomous University of Mexico. Rats were housed under controlled temperature and humidity on a 12 h light cycle (lights on at 07:00, off at 19:00) with ad libitum access to standard chow and water. All procedures were approved by the School of Medicine’s Ethics Committee (License: CIEFM-079-2020, approved on 1 February 2022 and approved for extension to 31 December 2026 on 27 March 2025).

### 4.1. Heart Beats and Blood Pressure Measurements

The blood pressure (BP) measurements in conscious rats (*n* = 9) were done between 10:00 a.m. and 03:00 p.m. via a non-invasive tail-cuff CODA system (Kent Scientific Corp., Torrington, CT, USA). Animals were acclimated to the restraining device and the sensing tail-cuff during 30 min (5 consecutive days) to avoid unnecessary stress in the course of measurements. On the day of measurements, rats were acclimated for 20 min before starting the recording of blood pressure. During recording, the occlusion cuff and the volume pressure recording (VPR) sensor were positioned on the tail base while the rats were maintained in a heater platform set to 30 °C to induce vasodilation, which enhances the blood flow to the tail. Data of BP and heart rate (the average of 3 times) were evaluated for each rat at the indicated time: (i) two times at basal level (indicated as −10 min and time 0); (ii) at 5, 10, 15, 30, 60, 90 and 120 min after hydralazine (10 mg/kg) or saline (0.9% NaCl, 1 mL/kg) intraperitoneal administration. CODA v2.0. software displays the systolic, diastolic and mean blood pressure data in mmHg units and heartbeats per minute. Data are expressed as the mean ± SEM of BP measurements in mmHg of at least 3 rats per condition and indicated time. Statistical analysis was performed using two-way ANOVA followed by Fisher’s LSD multiple comparisons post hoc test; a statistically significant difference was set with a *p* value < 0.05.

### 4.2. Immunohistochemistry for Light Microscopy

Adult male rats (*n* = 30) were perfused via the ascending aorta with 0.9% NaCl followed by 4% paraformaldehyde. Brains were sectioned at 70 µm using a Leica VT1200 vibratome (Wetzlar, Germany). Sections were washed in phosphate buffer (PB), blocked for 2 h in TBST (0.05 M Tris, 0.9% NaCl, 0.3% Triton-X) plus 10% normal donkey serum, and incubated overnight at 4 °C with primary antibodies (see Table 1) diluted in TBST with 1% serum. After washing, sections were incubated for 2 h at room temperature with appropriate biotinylated or fluorescent secondary antibodies (1:500; Vector Laboratories, Newark, CA, USA). Brightfield images were acquired using a Nikon Eclipse E600 (Tokyo, Japan) and confocal images were acquired with Zeiss LSM 880 (Jena, Germany).

### 4.3. Fos Counting

Regions of interest (ROIs) were delineated on bright-field images using Swanson atlas landmarks (KF, CeC, BSTov, PVN subnuclei, SON). Within each ROI, non-overlapping fields of 60 µm × 53 µm (0.0032 mm^2^) were sampled bilaterally and quantified as cells per field. A neuron was scored Fos immunopositive only when it contained a discrete intranuclear DAB deposit fully enclosed by the hematoxylin counterstain, with round/oval morphology and intensity clearly above local background; cytoplasmic staining, diffuse puncta, and glial-like profiles were excluded. Unbiased counting-frame rules were applied (top/left borders inclusive; bottom/right borders exclusive) to avoid edge effects; fields with tissue folds, tears, or vascular artifacts were omitted. For each animal, all fields from all available sections within the defined atlas levels were averaged to yield a single per-animal value for each ROI; these per-animal means were used for statistics. Counting was performed blind to group identity, with identical illumination and threshold settings maintained across groups for a given staining batch.

### 4.4. Statistics for Fos Counts

We analyzed per-animal means using a two-way ANOVA with factors Region (KF, CeC, BSTov, PVN lmd/ps/mpd, SON) and Group (Sleeping, Saline 60′, HDZ 60′, HDZ 120′). When the Group main effect or Region × Group interaction was significant, we probed simple effects within each region by one-way ANOVA followed by Tukey’s HSD pairwise comparisons (α = 0.05, two-tailed). In bar graphs, different letters above bars indicate groups that differ significantly within that region. Data are shown as the mean ± SEM; the animal is the statistical unit. Normality was checked; no corrections were required.

### 4.5. RNAscope In Situ Hybridization

Adult rat brains (*n* = 6, 3 saline, 3 HDZ) were rapidly dissected, flash-frozen, and stored at −80 °C. Coronal sections (12 µm) were cut at −20 °C and mounted onto slides. RNAscope was performed using manufacturer instructions (Advanced Cell Diagnostics, Inc, ACDbio.com, Newark, CA, USA). Sections were fixed, dehydrated, and treated with Protease IV before hybridization with target probes at 40 °C for 2 h. The following probes purchased from Advanced Cell Diagnostics, Inc, (ACDbio.com) were used: *Adcyap1* (channel 1, e.g., Figure 4A, Cat No. 400651-T1, channel 2, e.g., Figure 4C–F, Cat No. 400651-T2), *Slc17a7* (Cat No. 317001-T1), *fos* (Cat No. 403591-T1) and *Calca* (Cat No. 317518). After hybridization, slides were washed, counterstained with Gill’s Hematoxylin I (Sigma-Aldrich, diluted 1:2, St. Louis, MO, USA), and coverslipped.

### 4.6. Juxtacellular Labeling and Histological Analysis

Juxtacellular recordings were performed in 14 adult male rats under urethane anesthesia (1.5 g/kg, i.p.). Glass micropipettes (tip resistance 25–40 MΩ, WPI, Sarasota, FL, USA) were filled with 1.5% Neurobiotin (Vector Laboratories) dissolved in 0.15 M NaCl. Neurons were recorded extracellularly in the Kölliker–Fuse (KF) region of the parabrachial nucleus, identified by stereotaxic coordinates (−9.2 mm posterior to bregma, 2.9 mm lateral, and 7.5 mm ventral from the brain surface). Following electrophysiological identification, single neurons were labelled by juxtacellular iontophoresis of Neurobiotin using positive current pulses (200 ms, 1–10 nA, 50% duty cycle) applied for 5 min, once modulation of the firing rate was observed. After a survival period of 6 h, the animal was deeply anesthetized and perfused transcardially with 0.9% NaCl, followed by 4% paraformaldehyde in PBS + 15% *v*/*v* of a saturated solution of picric acid. Brains were sectioned sagitally at 70 µm using a vibrating microtome. Free-floating sections were processed first with Streptavidin coupled to Alexa 488, and then turned into a permanent DAB labelling using an avidin-biotin–peroxidase complex (ABC Elite, Vector Laboratories) and 3,3′-diaminobenzidine (DAB) as chromogen. To reveal co-localization of the labelled axons with pituitary adenylate cyclase-activating polypeptide (PACAP), some sections with labeled axonal profiles within the amygdala were incubated with Streptavidin-488 together with a rabbit anti-PACAP antibody (1:1000; BMA, cat. T-4473), followed by a fluorophore-conjugated secondary antibody. Sections were mounted and coverslipped. Labeled neurons and their projections were imaged using brightfield and epifluorescence microscopy, and a reconstruction of their location, as well as of their soma, dendrites and axonal projections, was performed using a camera lucida, coupled to a microscope. Axonal trajectories were reconstructed through the dorsal tegmental bundle and ansa peduncularis to the bed nucleus of the stria terminalis (BSTov) and amygdala. Digital micrographs were adjusted for brightness and contrast only.

## 5. Conclusions

We define a KF PACAP/VGluT1 projection that forms calyceal terminals onto PKCδ^+^/GluD1^+^ neurons in the extended amygdala and is selectively engaged during the early phase of recovery from acute hypotension. This pathway provides a mechanistic substrate for rapid autonomic compensation that precedes and complements slower vasopressin-dependent endocrine support, revealing a temporally stratified architecture for cardiovascular homeostasis.

## Figures and Tables

**Figure 1 ijms-26-11405-f001:**
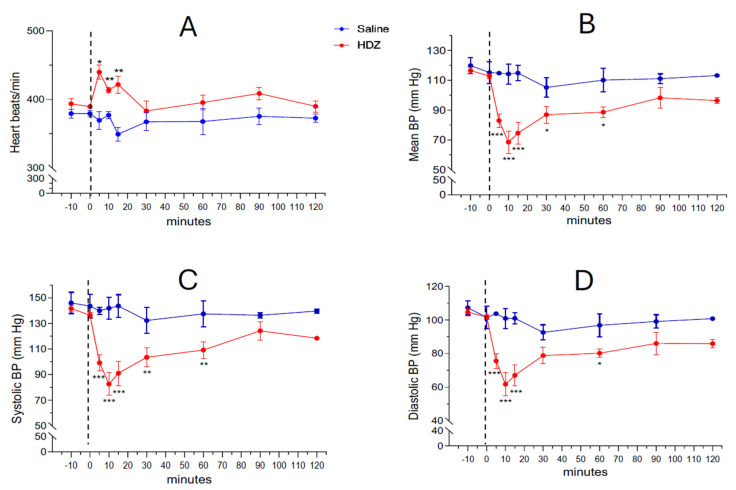
Effect of acute systemic hydralazine (HDZ) administration on heart rate and blood pressure. (**A**) Heart rate (beats per minute) following intraperitoneal injection of HDZ (red) or saline (blue) at time 0 (symbolized by the dashed line). (**B**) Time course of mean arterial blood pressure (BP) recorded from baseline (−10 and 0 min) to 120 min after injection. (**C**,**D**) Time courses of systolic (**C**) and diastolic (**D**) BP changes, respectively. Data are expressed as the mean ± SEM (HDZ, *n* = 6 rats; saline, *n* = 3 rats). * *p* < 0.05; ** *p* < 0.01; *** *p* < 0.001, two-way ANOVA followed by Fisher’s LSD multiple comparisons test.

**Figure 2 ijms-26-11405-f002:**
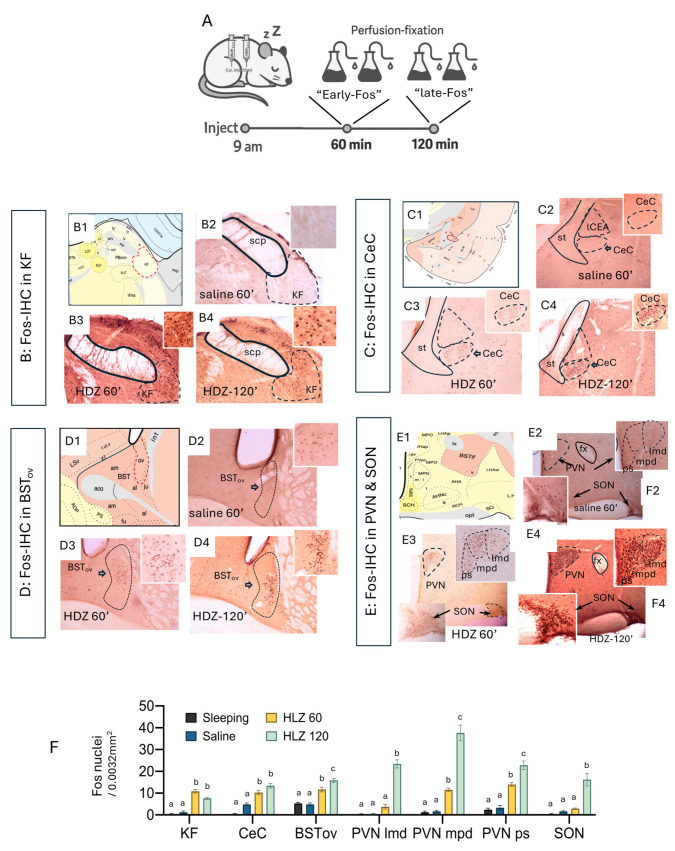
Sequential Fos activation in pontine KF, extended amygdala, and hypothalamic PVN/SON during recovery from acute hypotension. (**A**) Experimental design. Rats were assigned to four groups and perfused at the indicated times after a 9:00 i.p. injection: (i) Sleeping baseline (pentobarbital anesthesia followed by immediate perfusion), (ii) Saline (0.9% NaCl, 1 mL/kg) at 60 min, (iii) HDZ 60′ (hydralazine, 10 mg/kg in 1 mL/kg saline) at 60 min, and (iv) HDZ 120′ at 120 min. (**B**–**E**) Fos immunohistochemistry (IHC) across regions. For each block, panel 1 shows the corresponding Swanson atlas level; panel 2 shows Saline 60′; panel 3 shows HDZ 60′; panel 4 shows HDZ 120′. (**B1**–**B4**) KF (Kölliker–Fuse nucleus, ventrolateral parabrachial complex). (**C1**–**C4**) CeC (capsular central amygdala). (**D1**–**D4**) BSTov (oval nucleus of the bed nucleus of the stria terminalis). (**E1**–**E4**) PVN and SON (paraventricular and supraoptic nuclei of the hypothalamus). Insets illustrate nuclear Fos at higher magnification. (**F**) Quantification. Fos-positive nuclei (mean ± SEM) are plotted as cells per 0.0032 mm^2^ (60 µm × 53 µm field) for each region and group (Sleeping, Saline 60′, HDZ 60′, HDZ 120′). Different letters above bars indicate significant differences (two-way ANOVA within region followed by Tukey post hoc, α = 0.05). See Methods for sample sizes and counting criteria. PVN (paraventricular nucleus) subdivisions: lmd: lateral magnocellular division, mpd: medial parvocellular division, ps: presympathetic division.

**Figure 3 ijms-26-11405-f003:**
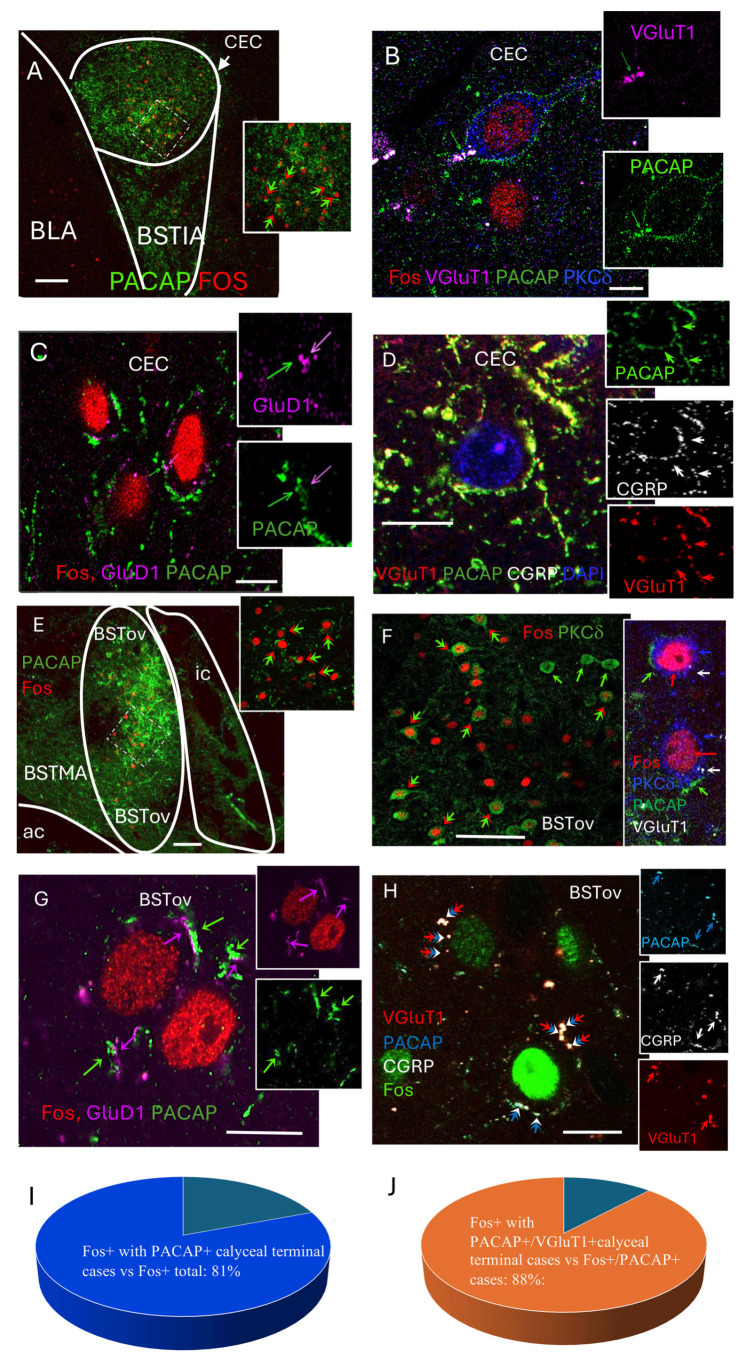
Early-phase Fos expression reveals selective activation of calyceal synapse–targeted neurons in the central amygdala (capsular division, CeC) and the oval subdivision of the bed nucleus of the stria terminalis (BSTov). (**A**) Low-magnification micrograph showing Fos-expressing neurons in the CeC. Inset: enlarged region from the boxed area in the main panel illustrating Fos-positive nuclei surrounded by PACAP-immunoreactive (ir) ring-like structures (bi-color arrows). (**B**–**D**) Thin confocal sections (1 Airy unit; 1 µm optical thickness) showing: (**B**) a PKCδ-immunoreactive (blue) neuron expressing Fos (red) at the early activation phase and contacted by PACAP/VGluT1 co-expressing calyceal terminals (green/magenta); (**C**) PACAP-immunoreactive axons opposed by GluD1-positive puncta on the cytoplasmic side; and (**D**) PACAP-immunoreactive calyceal terminals co-expressing CGRP and VGluT1. (**E**) Low-magnification micrograph showing Fos-expressing neurons in the BSTov. Inset: enlarged region from the boxed area illustrating Fos-positive nuclei surrounded by PACAP-immunoreactive ring-like structures (bi-color arrows). (**F**) Low-magnification image showing Fos-expressing neurons co-expressing PKCδ (green; indicated by bi-color arrows). Inset: two Fos/PKCδ^+^ neurons enveloped by PACAP/VGluT1-immunopositive calyceal terminals. (**G**) Thin confocal section (1 µm) showing GluD1-immunoreactive puncta apposed to PACAP-positive axons in the BSTov. (**H**) Thin confocal section (1 µm) showing VGluT1/CGRP/PACAP co-expressing axon segments surrounding Fos-positive nuclei. (**I**,**J**) Quantitative summaries: in panel (**I**) the proportion of Fos-positive neurons surrounded by PACAP-positive calyceal terminals (42 of 52 cases; 80.8%, represented with the light blue segment), while dark blue segment represents the proportion of Fos+ cells without observed PACAP + calyceal terminals. Panel (**J**) shows the proportion of Fos-positive neurons contacted by PACAP/VGluT1-positive calyceal terminals relative to Fos/PACAP-positive cases (37 of 42 cases; 88.1%, represented in the orange segment), while dark blue segment represents the proportion of the non-overlapping cases. Scale bars: (**A**,**E**,**F**) 100 µm; (**B**–**D**,**G**,**H**) 10 µm.

**Figure 4 ijms-26-11405-f004:**
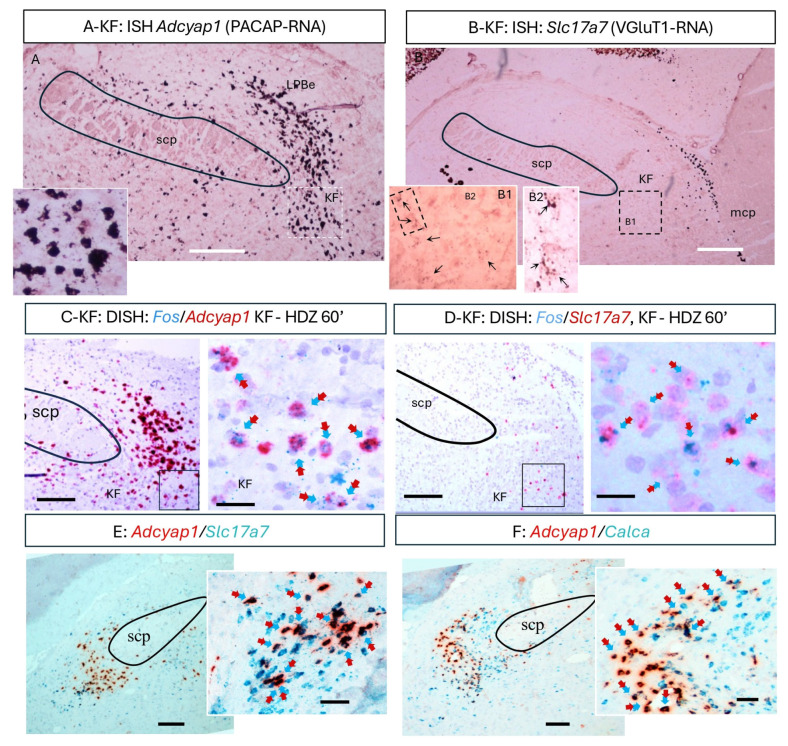
A subpopulation in the pontine Kölliker–Fuse nuclei (KF), which expressed mRNA of PACAP/VGluT1, engaged in the early phase of the acute hypotension event. (**A**,**B**) ISH using high sensitivity/low background single cell labeling with RNAscope brown-kit method revealed a peculiar cell population of rat brain located in the ventral subdivision of the pontine parabrachial complex, at bregma −9.12 mm to −9.24 mm, medio-lateral 2.90–3.00 mm and dorso-ventral 7.40 mm to 7.60 mm, between the superior and medial cerebellar peduncles (SCP and MCP, respectively), with large cell bodies, expressing *Adcyap1* (mRNA for PACAP, panel (**A**) and inset) and *Slc17a7* (mRNAs for VGluT1, panel (**B**) and inset, where the *Slc17a7* positive grains can be clearly seen on and surround the Nissl stained nuclei, black arrows). (**C**,**D**) With the RNAscope duplex method (DISH), it is evident that this KF^PACAP/VGluT1^ population is fos positive at the early raising phase. (**E**,**F**) DISH reactions show the co-expression of *Adcyap1* (PACAP) co-expressing *Slc17a7* (VGluT1) and Calca (CGRP). Scale bars: (**A**,**B**) 200 µm, (**C**–**F**): left panels, 50 µm, right panels: 25 µm.

**Figure 5 ijms-26-11405-f005:**
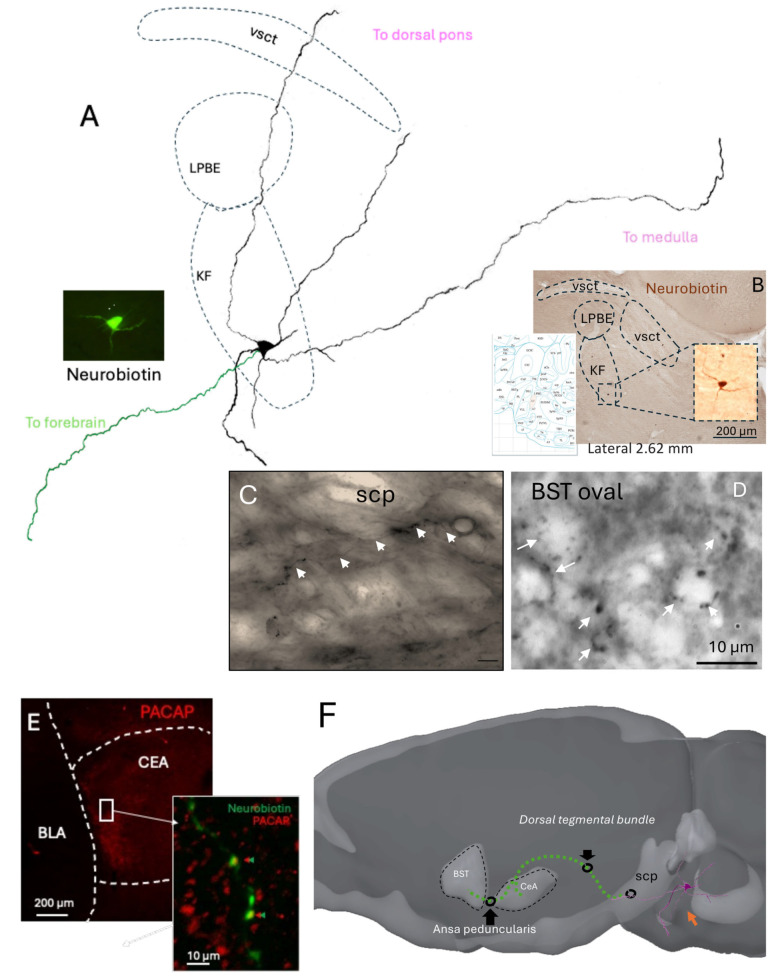
A juxtacellularly labeled KF neuron with long-range projections to medulla, dorsal pons, and forebrain targets. (**A**) Camera lucida reconstruction. Ten serial 60 µm sections were aligned to reconstruct a single Kölliker—Fuse (KF) neuron filled juxtacellularly with neurobiotin. The axon issues three major branches coursing medially/caudally to the medulla, dorsally within the pons, and rostrally toward the forebrain. Local landmarks are outlined (LPBe, external division of the lateral parabrachial nucleus; vsct, ventral spinocerebellar tract). (**B**) Soma location. Bright-field micrograph showing the labeled KF neuron at its origin within the ventrolateral parabrachial complex (inset: higher magnification of the soma). Scale bar, 200 µm. (**C**) Axon within the SCP. Neurobiotin-labeled axon (arrows) traveling in the superior cerebellar peduncle (SCP). Scale bar, 10 µm. (**D**) Ring-like terminal in BSTov. Enlarged neurobiotin-positive perisomatic “calyceal-like” terminal in the oval BNST (BSTov). Scale bar, 10 µm. (**E**) Forebrain terminals co-localize with PACAP. Left: PACAP immunofluorescence (red) in the capsular central amygdala (CeC) and adjacent basolateral amygdala (BLA). Right inset: merged confocal image showing neurobiotin-labeled terminals (green) apposed to PACAP-immunoreactive perisomatic profiles (red) in the CeC. Scale bars: left 200 µm, inset 10 µm. (**F**) Schematic (sagittal). Summary of the reconstructed trajectory: the rostrally directed branch enters major brain conduction systems, SCP to dorsal tegmental bundle (DTB) to CeC, and continues via the ansa peduncularis to the BSTov; additional branches descend toward the medulla. This single-cell anatomy supports a KF origin for PACAP-positive calyceal innervation of the extended amygdala. Abbreviations: BLA, basolateral amygdala; CeC, capsular central amygdala; DTB, dorsal tegmental bundle; KF, Kölliker—Fuse nucleus; LPBe, lateral. CeA: central amygdala.

**Table 1 ijms-26-11405-t001:** Primary antibodies used in this study.

Target	Host Species	Dilution	Source	Cat. Number
PACAP	Rabbit	1:2000	BMA Biomedical, Augst, Switzerland	T-4473
PACAP	Mouse	1:2000	Hannibal lab (Copenhagen, Denmark) [43]	MabJHH1
VGluT1	Guinea pig	1:5000	Chemicon (Tokyo, Japan)	AB-5905
CGRP	Rabbit	1:1000	Phoenix Pharmaceuticals (Burlingame, CA, USA)	H-015-09
CGRP	Goat	1:000	Abcam (Cambridge, UK)	Ab36001
CGRP	Mouse	1:1000	Abcam	Ab81887
Fos	Rabbit	1:2000	EnCor (Gainesville, FL, USA)	RPCA-c-FOS
Fos	Mouse	1:1000	Santa Cruz Biotechnology (Dallas, TX, USA)	sc-271243
Fos	Goat	1:1000	Santa Cruz Biotechnology	sc-253
GluD1	Guinea Pig	1:1000	Frontier Institute (Hokkaido, Japan)	GluD1C-GP-Af860
PKCδ	Mouse	1:1000	BD-Transduction Labs (San Jose, CA, USA)	610397

## Data Availability

The original contributions presented in this study are included in the article. Further inquiries can be directed to the corresponding author.

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
