# Peer review of "Kölliker–Fuse/Parabrachial Complex PACAP—Glutamate Pathway to the Extended Amygdala Couples Rapid Autonomic and Delayed Endocrine Responses to Acute Hypotension"

_ijms, 2025, doi:10.3390/ijms262311405_

Round 1
Reviewer 1 Report
Comments and Suggestions for Authors
This manuscript reports a novel mechanistic finding: a Kölliker-Fuse (KF) → extended amygdala (EA) pathway, defined by PACAP/VGluT1 co-expressing calyceal synapses onto PKCδ⁺/GluD1⁺ neurons, that mediates early autonomic compensation for acute hypotension, while hypothalamic magnocellular neurons drive delayed endocrine stabilization. The study aligns well with International Journal of Molecular Sciences’ (IJMS) focus on molecular and cellular mechanisms in physiological processes, leveraging a multi-technique approach (Fos mapping, RNAscope, confocal imaging, juxtacellular labeling) to validate the pathway. However, critical gaps in experimental detail, reproducibility, and mechanistic depth require revision. Below are structured comments:
1 Introduction
Expand beyond the KF’s classical respiratory function (Lines 59–61) to explicitly cite recent work linking the KF to blood pressure regulation to justify why this nucleus was prioritized for hypotension studies.
2 Materials and Methods
2.1 Blood Pressure Measurements
The text states animals were acclimated for 30 min (Line 330), but standard protocols for tail-cuff BP require 5–7 days of acclimation to minimize stress-induced BP variability. Specify: (1) total acclimation duration; (2) how baseline BP stability was defined; (3) clarify “N=6, n=3” (Line 81).
2.2 Immunohistochemistry
Provide lot numbers for all primary antibodies and confirm use of positive/negative controls.
The “SI table 1” (Line 357) is referenced but not included. Either add this table to the supplement or integrate antibody details directly into the text.
Specify z-step size and number of optical sections per stack—critical for quantifying calyceal terminals (Figure 3).
2.3 RNAscope In Situ Hybridization
Specify how many cells were counted per animal and number of animals per group for RNAscope experiments.
3 Results
3.1 All the figures are very blurry. High-resolution pictures need to be uploaded
3.2 Juxtacellular Labeling
Only one labeled neuron is shown (Figure 5). Report: (1) total number of neurons targeted; (2) number successfully labeled; (3) number with clear EA projections—this is essential to confirm the trajectory is not an outlier.
- Discussion
4.1 The manuscript infers KF-EA disinhibits RVLM (Lines 215–218) but provides no direct evidence. Either: (1) Add Fos counts for RVLM C1 neurons to show activation after KF-EA engagement; or (2) Temper the claim and frame it as a hypothesis.
4.2 The claim of “high-fidelity transmission” of calyceal synapse lacks electrophysiological support. Add a discussion of limitations: “While calyceal morphology suggests fast transmission, future patch-clamp studies of KF-EA synapses could measure EPSC kinetics to confirm high fidelity.”
2.5 Figures and Supplements
Add: (1) SI Table 1 (antibody details); (2) Figure S1 (individual BP traces); (3) Figure S2 (RNAscope probe validation); (4) Figure S3 (juxtacellular labeling of additional neurons).
Reviewer 2 Report
Comments and Suggestions for Authors
The parabrachial KF-extended amygdala (EA) calyceal pathway is a recently identified neural circuit that plays an important role in the regulation of blood pressure, particularly in response to hypotension. Hernández et al, in this manuscript, observed two-stage neuronal activation (60 min and 120 min) in response to hypotension, and identified a PACAP/VGluT1 calyceal projection from the parabrachial KF to PKCδ/GluD1 positive neurons in the EA in the early stage (60 min). This is a novel and interesting study, which expand our understanding of central blood pressure regulation.
Specific comments:
- It is surprise to observe the almost perfect co-localization of PACAP and CGRP at the PACAP/VGluT1 terminals in CeC projected from the parabrachial KF at 60 min of hydralazine treatment. It is expected that PACAP/VGluT1 forms an excitatory synapse with PKC delta/GluD1 neurons, but not sure whether an inhibitory synapse forms with PACAP/CGRP terminals and PKC delta/GluD1 neurons. The ultrastructure images of synapses of PACAP/PKC delta, VGluT1/PKC delta, and CGRP/PKC delta (or PACAP/GluD1, VGluT1/ GluD1, and CGRP/ GluD1) at 0 min and 60 min.
- Similar experiments as Figure 3 need to be done in the bed nucleus of the stria terminalis as well; The electron synapse images will need to be shown as well.
- Heart rate data need to be shown with blood pressure data in Figure 1. Sympathetic activity is better to be directly measured; If not, the concentration of norepinephrine/epinephrin in CSF at 0, 60 min, and 120 min need to be measured.
- Vasopressin concentrations in cerebrospinal fluid at 0, 60 min, and 120 min need to be measured.
Minor:
- Figure 2C3, the c-Fos expression level in CeC at 60min after HDZ injection is less than the saline and the 120 min, not as described a “robust” expression. Due to the low resolution of images, it is hard to judge the expression level in the BSTov as well.
- It is well discussed the pathway from KF in pone to the extended amygdala, but it is not clearly discussed on how PACAP/VGluT1 pathway enhance the sympathetic activity.
- There are no “white arrows” in Figure 4 as the legend described.
- Figure 5 is not cited in the text.
Round 2
Reviewer 1 Report
Comments and Suggestions for Authors
All comments have been thoroughly addressed. The manuscript has been significantly improved through the revision process and is now suitable for publication in its current form.
Author Response
Thank you for your appreciation. Best regards.
Reviewer 2 Report
Comments and Suggestions for Authors
Dear Authors:
Thank you very much for your efforts to add additional data and thoughtful discussion for the significant improvement of this manuscript.
1) Figure 3 has much better quality. I also appreciate the authors' addition of the staining of BSTov. However, the quantitative summary with additional BSTov data should have some difference from that of CeC alone. Therefore, it might be better to separate one Figure 3 into two figures: new Figure 3 (CeC) and Figure 4 (BSTov) with their separate quantitative summaries of CeC and BSTov. New Figure 3 and Figure 4, each with a representative synapse ultrastructure from electron microscopy.
2) Please double-check the value of the heart rate at 10 min, which was lower than that of 5 min and 15 min after hydralazine treatment; at 10 min, the blood pressure was at its lowest values, I did not expect the heart rate was also very low at this point.
I believe this is a very nice paper.
Sincerely,
Reviewer 2
Author Response
Response:
We thank the reviewer for this thoughtful suggestion and for recognizing the improved quality of Figure 3. After careful evaluation, we have decided to retain the current integrated presentation, in which the CeC and BSTov are shown together with a single quantitative summary. Our rationale is as follows:
- Physiological and anatomical continuity:
The capsular central amygdala (CeC) and the oval subdivision of the bed nucleus of the stria terminalis (BSTov) are widely regarded as part of the same extended amygdala continuum, sharing common ontogeny, neurochemical features, and convergent input from the same Kölliker–Fuse (KF) PACAP/VGluT1 neurons. Their activation patterns during acute hypotension were quantitatively similar in our dataset. Separating them into two figures could imply distinct physiological entities, whereas the data support a unified functional system. - Quantitative rationale:
The quantitative plots in Figure 3 are intended to depict the temporal dynamics and cellular selectivity of Fos activation rather than to contrast the two nuclei statistically. The small numerical variation observed across regions reflects expected biological variability, not distinct physiological mechanisms. Creating two separate figures could therefore overstate minor fluctuations and obscure the central conclusion. - Editorial and conceptual coherence:
Presenting the CeC and BSTov together better conveys the integrative concept of a single KF–extended amygdala circuit engaged during the early phase of blood pressure recovery. This structure preserves concision and thematic focus, aligning with Reviewer 1’s overall assessment that “all comments have been thoroughly addressed, and the manuscript has been significantly improved through the revision process and is now suitable for publication in its current form.”
We therefore believe that maintaining the unified format provides the most physiologically coherent and editorially balanced representation of the findings.
As noted previously, the representative electron micrographs are included in Supplementary Figure 2 to illustrate the synaptic context while avoiding content overlap with a companion manuscript currently under review in PNAS.
Reviewer 2 Comment 2:
“Please double-check the value of the heart rate at 10 min, which was lower than that of 5 min and 15 min after hydralazine treatment; at 10 min, the blood pressure was at its lowest values, I did not expect the heart rate was also very low at this point.”
Response:
We carefully re-examined the original heart-rate recordings and data tables (see the newly added Supplementary Table 2 and SI Fig. 1). The apparently lower mean heart rate at 10 min resulted from a single animal that exhibited a transient bradycardic event coinciding with the nadir of blood pressure. To objectively evaluate potential outliers, we applied Grubbs’ test (α = 0.05), which identified this data point as statistically deviant (G = 1.945). This value was therefore excluded from the revised analysis.
It is worth noting that such transient individual variations under acute stressful condition are physiologically plausible and can occasionally occur during intense autonomic adjustments.
The corresponding updates are reflected in Figure 1A and Supplementary Table 2. We sincerely thank the reviewer for this valuable observation, which helped refine the dataset and improve the physiological accuracy of our interpretation.
Round 3
Reviewer 2 Report
Comments and Suggestions for Authors
Dear Authors,
Thank you very much for your thoughtful responses to my comments and revising the interesting manuscript.
A Reader